

# Nonconventional opponents: a review of malaria and leishmaniasis among United States Armed Forces

Kaylin J. Beiter, Zachariah J. Wentlent, Adrian R. Hamouda and Bolaji N. Thomas

Department of Biomedical Sciences, College of Health Sciences and Technology, Rochester Institute of Technology, Rochester, NY, United States of America

## ABSTRACT

As the United States military engage with different countries and cultures throughout the world, personnel become exposed to new biospheres as well. There are many infectious pathogens that are not endemic to the US, but two of particular importance are *Plasmodium* and *Leishmania*, which respectively cause malaria and leishmaniasis. These parasites are both known to cause significant disease burden in their endemic locales, and thus pose a threat to military travelers. This review introduces readers to basic life cycle and disease mechanisms for each. Local and military epidemiology are described, as are the specific actions taken by the US military for prevention and treatment purposes. Complications of such measures with regard to human health are also discussed, including possible chemical toxicities. Additionally, poor recognition of these diseases upon an individual's return leading to complications and treatment delays in the United States are examined. Information about canine leishmaniasis, poorly studied relative to its human manifestation, but of importance due to the utilization of dogs in military endeavors is presented. Future implications for the American healthcare system regarding malaria and leishmaniasis are also presented.

Corresponding author
Bolaji N. Thomas, bntsbi@rit.edu

## INTRODUCTION

International deployment is a common occurrence in the United States (US) Armed Forces, comprising of the Marine Corps, Navy, Air Force and Army, with nearly 200,000 troops currently stationed overseas (*Bialik, 2017*). Leaving the US and its native biosphere for their posted sites introduces a risk of exposure to, and infection with, endemic parasitic, bacterial, viral or fungal diseases (*Ciminera & Brundage, 2007*). Such hazards are well-known; during World War I, approximately half a million allied soldiers were diagnosed with malaria in just one year (*Beaumier et al., 2013*; *Brabin, 2014*). As the United States alters its current foreign stance, engaging with new regions in the process, deployment locations shift accordingly, with infectious disease risks paralleling the revisions in political alignment. Most recently, US troops have been highly clustered in the Middle East and Africa, specifically Iraq and Afghanistan, with sprinkling of Special Forces operatives in Syria. Endemic diseases in this Middle Eastern region are caused by an amalgam of local viruses, parasites, and bacteria, with the most clinically relevant including Middle East Respiratory Syndrome (MERS),

dengue fever, schistosomiasis, enterotoxigenic *Escherichia coli* (ETEC), leishmaniasis, brucellosis, and toxoplasmosis (*Hotez, Savioli & Fenwick, 2012*; *Humphrey et al., 2016*; *Raad et al., 2018*; *Foster et al., 2018*). These pathogens, beyond disease complications, have significantly impacted local economic development, limited infrastructure, and reduced life expectancy of nationals causing an ever-increasing reliance on pharmaceuticals (*Hui et al., 2018*; *Park et al., 2018*). Consequently, these same pathogens have negatively affected members of the US military stationed at such locations (*Faulde et al., 2008*; *Fukuda et al., 2011*), both with regard to direct infection of troops as well as limitations in treatment accessibility.

High infection rates of parasitic diseases, principally malaria and leishmaniasis, are expected in these locales, and the deployed members of the United States military have not been spared (*Mitchell, Silvitz & Black, 2007*; *Mace, Arguin & Tan, 2018*; *O'Donnell, Stahlman & Fan, 2018*), with infectious diseases proving to have devastating effects on the military, even contemporarily (*Armed Forces Health Surveillance Branch, 2018*). Since initiation of the activities in the Middle East, injuries sustained in battle were six times less common than those caused by non-battle injuries (i.e., infection). However, the possibility for disease is not limited to deployed troops. The risk of bioterrorism, domestic or foreign, from increasingly radical groups is growing all over the world (*Anderson et al., 2005*; *Beaumier et al., 2013*), leading to the expansion of disease prevention and force protection strategies by the Department of Defense. These efforts are subject to budgetary restrictions and cuts to research funding, despite overall increase in military spending, and could have serious health implications beyond the United States military (*Beaumier et al., 2013*).

Despite the vast number of clinically relevant pathogens in locales where members of the US military are deployed, this review will address only two of such infections: malaria and leishmaniasis. The seriousness of these diseases and their implications for healthcare system within the homeland demands critical analysis, and form the basis for this review. Due to the chronic and sometimes relapsing nature of these diseases, infected individuals may present symptoms months/years after exposure (*Goodrich et al., 2017*; *Nagarajan & Sloan, 2015*), especially following subsequent immune system compromise (*Mansueto et al., 2014*). Possible re-emergence of pathogenic forms in such individuals allows them to serve as agents of autochthonous infections at home.

## Survey methodology

For the purpose of this review, we divided the article into sections and disease-specific subsections, focusing on specific diseases, one at a time. To access relevant and related publications, we carried out a search on pubmed.gov for armed forces deployment information, using AND as the link word, as the case may be. For others, we searched journal specific or government websites (http://www.health.mil), in addition to websites of international organizations such as the World Health Organization (https://www.who.int/tdr/en/) and the Centers for Disease Control (http://www.cdc.gov), to retrieve articles focused on parasitic diseases in specific endemic locations. Subject directed keywords or terms utilized in our search include armed forces, deployment, malaria, leishmaniasis, epidemiology, military-related autochthonous parasitic diseases,

prevention and treatment. References include articles directly relating to research data, relevant case reports or clinical presentations in the United States, Canada, or Europe from deployed service men and women. Of additional interest in our search were articles relating to zoonotic (canine) leishmaniasis among military dogs deployed overseas alongside their handlers and potential for disease transmission on return to the United States.

## Malaria

Despite clinical notoriety and insidiousness of more 100 years, the availability of modern preventative measures and a better understanding of disease dynamics, the worldwide threat of malaria in endemic countries and menace among deployed members of the armed forces remains. At its apex in the Middle East, the United States Armed Forces suffered an incident rate of 52.4 cases per 1,000 troops (*Beaumier et al., 2013*). Of recent, there has been a significant reduction in the number of troops stationed in these locations and the implementation of evidence-based preventative measures against malaria. Despite this, an update from 2017 reveal 32 cases of malaria infection among US military stationed overseas, with cases reported from facilities as far as Afghanistan, Korea, Japan, Djibouti (*Armed Forces Health Surveillance Branch, 2018*). Though the percentage of individuals affected was not given, the total reduction in malaria cases in recent years has been ascribed to US troops leaving the Middle East region (*Armed Forces Health Surveillance Branch, 2018*). This number however does not include self-limited or subclinical cases, and the true attack rate may be higher than published.

### *Causes*

Human malaria is spread via bloodmeals of the female *Anopheles* mosquito, which picks up gametocytes while feeding on infected individuals, and hosts the developmental stages until full maturation into infective sporozoites, which are then passed on to a new host during another mosquito feeding process (*Anderson et al., 2005*). Clinical malaria is classically due to any of four different *Plasmodium (P)* species: *P. vivax, P. falciparum, P. ovale and P. malariae,* though an animal species, *P. knowlesi* has been confirmed as responsible for significant human infections in parts of southeast Asia (*Muller & Schlangehauf, 2014*; *Millar & Cox-Singh, 2015*; *Divis et al., 2015*). The dominant species causing infection among the armed forces is *P. vivax* (*Ciminera & Brundage, 2007*), although the latest update on malaria in the military shows a shift to increasing infections with *P. falciparum,* as the causative agent for the most number of cases (*Armed Forces Health Surveillance Branch, 2018*), likely due to the locations of reporting. *P. vivax,* found primarily in Asia and Latin America, and in some parts of Africa, differs from other species in that its life cycle includes a dormant liver stage, during which transformed parasites (otherwise called hypnozoites), can remain for months or even years after the initial mosquito bite, becoming symptomatic if the parasites leave the liver to invade healthy red blood cells (*Baird et al., 2016*), and responsible for relapses. It is uniquely dangerous in that infected patients may remain subclinical for years after returning home from military service (*Kotwal et al., 2005*), serving as agents of future autochthonous infection. Furthermore, because malaria is no longer endemic in the United States, healthcare providers may see a patient presenting with symptoms similar to

flu symptoms, without suspecting malaria as the potential cause of such clinical symptoms. This leads to delays in diagnosis and institution of appropriate treatment.

The symptoms of *P. vivax* malaria are consistent with all types of malaria: fever, chills, nausea/vomiting, myalgia, fatigue, and general malaise (*Yohannes & Ketema, 2016*), with infection and lysis of red blood cells during the active erythrocytic cycle leading to jaundice and anemia (*Markus, 2011*). The liver hypnozoites do not all have an equal duration of senescence, and patients may experience paroxysmal symptomology, as parasites enter into active erythrocyte infection at different times (*Baird et al., 2016*). Such a vague presentation with common symptoms of general malaise often leads to misdiagnosis in the United States, where flu and other common ailments are worked up instead (*Evans et al., 2014*; *Goldman-Yassen et al., 2016*).

### Prevention and treatment

The United States military prioritizes prevention over treatment, implementing several protocols to this effect, which theoretically should make it almost impossible for service members to get malaria (*Kotwal et al., 2005*; *Shaha et al., 2013*). According to military documentation, there are five major malaria prevention strategies namely: (1) use of factory-treated uniforms; (2) regular application of $N,N$-diethyl-*meta*-toluamide (DEET) or picaridin to exposed skin; (3) proper wearing of military uniform; (4) use of permethrin-treated bed net, and (5) continuous chemoprophylaxis during all phases of deployment (*Robert, 2001*). Successful implementation of all five strategies however, is rare, limiting the efficacy of disease prevention. Furthermore, employment of such prophylactic measures, especially with mefloquine, subjects servicemen and women to significant adverse outcomes (*Adshead, 2014*).

Permethrin is an insecticide that kills mosquitoes by inducing spasms and paralysis (*Isaacs, Lynd & Donnelly, 2017*), while DEET, created by the United States Army in 1946, is less insecticidal, masking human scent to prevent mosquito bites (*Toynton et al., 2009*). Despite inclusion in malaria prevention programs, both chemicals have known associated dangers (*Diaz, 2016*). DEET is known to have heightened toxicity when applied with sunscreen (*Yiin, Tian & Hung, 2015*; *Rodriguez & Maibach, 2016*), a situation likely to arise for military personnel deployed to tropical and subtropical regions of the world. Additionally, synergism has been observed during simultaneous exposure to DEET and permethrin, leading to neuronal degeneration and significant neurobehavioral effects (*Abdel-Rahman et al., 2004*).

The chemoprophylactic drugs (doxycycline or mefloquine) administered by the military, also have known adverse effects, associated with use. Historically, mefloquine was developed by the US military (*Nevin, 2005*), but is now only considered if doxycycline is not tolerated. The side effects can be extreme, including psychiatric symptoms such as anxiety, paranoia, depression, hallucinations, and psychosis (*Tan et al., 2011*). Sequelae can also mimic post-traumatic stress disorder (PTSD), a condition for which deployed servicemen are at high risk (*Eick-Cost et al., 2017*). Doxycycline is by far safer and more reliable; side effects rare, and when present can range from nausea/vomiting and esophagitis to psychiatric symptoms like depression and anxiety (*Brisson & Brisson, 2012*).

Until recently, infection with *P. vivax* malaria was treated with primaquine for complete pathogen eradication. It acts by targeting the dormant hypnozoites in the liver, preventing the possibility for recurrence, thereby facilitating complete recovery (*Ashley, Recht & White, 2014*). Adverse effects have been documented in patients with rare preexisting genetic conditions such as glucose-6-phosphate dehydrogenase deficiency (*Valencia et al., 2016*; *Dombrowski et al., 2017*; *Watson et al., 2018*). Otherwise, the side effect profile is mild and comparable to all other anti-malarial drugs: nausea, vomiting, and abdominal cramps (*Burgoine, Bancone & Nosten, 2010*). Recently, tafenoquine was approved by the United States Food and Drug Administration for the radical cure of vivax malaria in patients aged 16 years and older (*Rajapaske, Rodrigo & Fernando, 2015*; *Tenoro, Green & Goyal, 2015*), though some genetic contraindications still exist. Both primaquine and tafenoquine should be made available for the treatment of United States military servicemen and women deployed overseas (*Watson et al., 2018*).

## Leishmaniasis

More than twenty *Leishmania* species have been identified, and the parasite is considered endemic throughout the world, including much of the Middle East (*Golding et al., 2015*). Since 2001, 2.4 million US troops have been deployed to Iraq and Afghanistan to participate in various military missions (*Spelman et al., 2012*). With a 2.1% contraction rate, leishmaniasis has become one of the most commonly diagnosed diseases since the commencement of military activities in both countries (*Beaumier et al., 2013*). As deployed personnel return home, US cases of leishmaniasis have risen to match levels seen during World War II (*Weina, Neafie & Wortmann, 2004*). Leishmaniasis is clinically relevant for armed forces and refugees alike, as civil conflict and unrest continues in the Middle East.

### *Causes*

The number of animal species which can serve as *Leishmania* reservoir hosts is ever increasing, including rodents, canines, and farm animals (*Stephens et al., 2016*). Leishmaniasis was not traditionally considered endemic to the United States, although recent epidemiologic findings reveal this status may be changing, secondary to globalization and autochthonous infections leading to persistent endemicity, especially along the southern border and new animal hosts (*Wright et al., 2008*; *Barry et al., 2013*; *McIlwee, Weis & Hosler, 2018*). Incidence and disease burden are higher in societies where people live in close proximity to host animals (*De Vries, Reedijk & Schallig, 2015*). Similar to malaria, it is a vector-borne disease, requiring a phlebotomine sand fly to pick up amastigotes during a bloodmeal from an infected (reservoir) host. Amastigotes undergoes development and maturity in the fly, which then inoculates infective promastigotes into a new mammalian host during the next blood meal. Disease manifests as three distinct clinical forms: cutaneous (including diffuse cutaneous form), mucocutaneous, or visceral. Cutaneous leishmaniasis (CL) has been the most diagnosed of the three among deployed United States servicemen and women (*Weil, 2010*; *Beaumier et al., 2013*), with *L. major,* the most prevalent (*Herwaldt, 1999*). Overall, 90% of global CL cases occur in Afghanistan, Brazil, Iran, Peru, Saudi Arabia, and Syria. With the United States military active involvement and troop deployment to

these locations, increased cases of leishmaniasis were recorded, until drawdown when the number of deployed soldiers reduced dramatically (*Shirian et al., 2013*; *De Vries, Reedijk & Schallig, 2015*). Expectedly, this is not unique to the US military, with cases of *L. major* infection and multiple reports of cutaneous disease among British, Dutch, and German soldiers as well (*Faulde et al., 2008*; *Bailey et al., 2012*).

## Symptoms

Each disease pattern has its own set of symptoms and thus differs in severity. Lesions can be self-resolving, as is often the case with many instances of CL, with or without resultant subclinical parasitemia (*Micallef & Azzopardi, 2014*; *Rosales-Chilama et al., 2015*; *Thomaidou et al., 2015*). The parasite can also disseminate into internal organs, as in visceral leishmaniasis, becoming fatal in the process. IL-12 and CD4+ Th1 cells have especially been implicated in the development of cellular immunity, though the mechanisms and explicit contributions of each are not yet fully understood (*Engwerda, Ato & Kaye, 2004*; *McCall, Zhang & Matlashewski, 2013*; *Buxbaum, 2015*; *Portela et al., 2018*). Fortunately, CL has been the dominant clinical form in the US military, though many subclinical cases, which do not require medical treatment occur as well (*Reithinger et al., 2007*). The infection first manifests as a simple, non-swollen, red ring around the bite from the sand fly. As the host immune system continues to respond locally at the bite site, sores develop on the skin and can further ulcerate, causing discomfort, but generally painless (*Reithinger et al., 2007*). In severe cases, these sores can develop on mucosal membranes, degrade the tissues of the mouth and tongue, and potentially interfere with swallowing or cause difficulty breathing. The incubation period for CL can range from 2 weeks to many months and even years, with potential delay between contraction of the disease and onset of symptoms. Soldiers may travel between countries, or even back to the United States, unwittingly becoming reservoir hosts in the process (*Nagarajan & Sloan, 2015*; *Goodrich et al., 2017*), leading to delay in diagnosis and institution of appropriate treatment, if symptoms occur after the patient has returned to a non-endemic area.

## Prevention and treatment

The incidence rate of leishmaniasis among the United States Armed Forces was 7.2 cases per 100,000 person-years for the period of 2001–2016, with the majority of cases being CL (*Stahlman, Williams & Taubman, 2017*). The reduced incidence rate of recent years has been attributed to better equipment and an emphasis on personal protective measures (*Rowland et al., 2015*). Nevertheless, there is still a cause for concern when troops are newly deployed to endemic regions; supportive resources may not be fully in place, deployed personnel may have limited knowledge, and a culture of preventative measure necessity may not have yet developed (*Coleman, Burkett & Putnam, 2006*). All of these factors can lead to an initial high caseload (*Oré et al., 2015*). The first line of protection from CL is through the use of personal protection techniques. However, there are no effective chemoprophylaxis drugs and no fully developed vaccines, and thus prevention of CL can be extremely difficult (*Ghorbani & Farhoudi, 2018*).

The preventative techniques that do exist focus on avoiding the bite of infected sandflies. Uniforms are impregnated with a type of pyrethroid (usually permethrin), insect repellants

are recommended, and personnel are given pyrethrin-treated bed nets (*Orsborne et al., 2016*). Theoretically, these measures should bring the incidence rate to near-zero levels, but this has not been the case so far. In 2003 alone, the incidence rate was estimated at 200 per 1,000 soldiers (*Gonzalez, Solís-Soto & Radon, 2017*). Efforts have also begun to be focused on reducing the population of sandflies, in order to mitigate transmission risk. Cyfluthrin, a pyrethroid insecticide, has been used to decimate sand fly populations, and chloropiricin, a wide-spectrum nematicide and insecticide, has been used to reduce the rodent (leishmania reservoir) population (*Aronson et al., 1998*; *Crum, 2005*).

Fortunately, modern medicine has afforded CL cure rates up to 91%. The most effective treatment is sodium stibogluconate, given intravenously at doses of 20 mg per kilogram of body weight, for 20 days (*Mitchell, Silvitz & Black, 2007*; *Stahlman, Williams & Taubman, 2017*), with side effects such as fatigue, arthralgia, myalgia, headaches, and chemical pancreatitis. Sodium stibogluconate is efficacious, but development of new drugs is imperative due to these side effect, the threat of drug resistance, and the high cost ($100 per 100 mL) (*Aronson et al., 1998*). The threat posed by recent reports of treatment failures in South and Latin America leishmaniasis cases, including the induction of transmissible skin microbiota that significantly promotes inflammation, should be a concern for all in the infectious disease community, particularly the military (*Mans et al., 2016*; *Obonaga et al., 2014*; *Gimblet et al., 2017*).

## Other: military zoonotic leishmaniasis

Canines are one of the main reservoirs for *Leishmania* species (*Burza, Croft & Boelaert, 2018*; *Quinnell & Courtenay, 2009*), with cases often subclinical. As the parasite multiplies in an asymptomatic dog, *Leishmania* is perpetuated locally via phlebotomine vectors (*Killian, 2007*). The usually implicated species, *L. infantum*, does not typically infect healthy humans, though incidence of associated infection and disease has increased in recent years (*Stoeckle et al., 2013*; *Kroidl et al., 2014*; *Bennai et al., 2018*; *Herrera et al., 2018*; *Risueno et al., 2018*; *Teimouri et al., 2018*), with immunodeficient individuals at a higher risk of disease (*Michel et al., 2011*). Zoonotic transmission of *L. infantum* to humans often results in visceral leishmaniasis infection (*Burza, Croft & Boelaert, 2018*). Though military personnel often have superior baseline health ratings compared to their civilian cohorts upon deployment, the stress of military life can contribute to development of an immunodeficiency state. Additionally, military personnel are at higher risks of smoking/alcohol/substance-abuse initiation and recidivism, frequently spend long periods of time in environments with sub-standard hygiene, may have more erratic sleep schedules, and overall suffer greater declines in mental and physical health (*Dau, Oda & Holodniy, 2009*; *Spelman et al., 2012*). In particular, personnel who have served in the Middle East since the Persian Gulf War have reported higher levels of psychosomatic/psychological pain in comparison to cohort military personnel that served contemporarily but in other locations (*Gray et al., 1996*; *Dlugosz et al., 1999*). Thus, troops are at a relatively high risk of zoonotic canine leishmaniasis due to exposure (military dogs becoming infected, local dogs in endemic areas) and high prevalence of immunodeficiency/extreme stress secondary to their service.

Canine Leishmaniasis (CanL) is a common veterinary problem worldwide, with recent prevalence estimates as high as 25–80% (*Michel et al., 2011*; *Akhtardanesh et al., 2017*; *Baneth et al., 2017*; *Guven et al., 2017*; *Ruh et al., 2017*; *Al-Bajalan et al., 2018*; *Monteiro et al., 2018*). Brazil is known to be especially affected (*Borja et al., 2016*; *Torres-Guerrero et al., 2017*; *Da Rocha et al., 2018*; *Melo et al., 2018*), with published reports highlighting increases in diagnoses and advocating for better public health strategies to be focused specifically on dogs (*Camargo & Langoni, 2006*; *Lima et al., 2010*). The armed forces regularly use military working dogs (MWD) for special operations, including abroad in *Leishmania*-endemic regions. CanL has been found multiple times in military animals (*Kawamura, Yoshikawa & Katakura, 2010*; *Davoust et al., 2013*). Of domestic importance, these dogs who are deemed 'adoptable' are mandated to return to the US, with adoption priority given to their former handlers. Approximately 1,000 former-MWDs enter the US annually (*Killian, 2007*), potentially serving as a reservoir for *Leishmania*: both symptomatic and asymptomatic canines have been shown to have similar inoculation abilities (*Moshfe et al., 2009*), and CanL prevalence has been shown to correlate with that of human disease (*Bruhn et al., 2018*). CanL was first found in the United States in the 1980s and 1990s infecting foxhounds, (*Enserink, 2000*; *Petersen, 2009*), with limited studies carried out since, to estimate current levels. Autochthonous infection has been reported since in North American dogs (*Schantz et al., 2005*). Without better screening for all MWDs, the possibility remains for import of CanL via former-MWDs and thus future augmentation in *L. infantum* incidence in US civilian and military populations. Canine vaccination, one possible solution to mitigating CanL disease burdens in the United States has shown some success as a human health preventative measure (*Palatnik-de Sousa et al., 2009*; *Rezvan & Moafi, 2015*; *Ribeiro et al., 2018*), though vaccine efficacy remains low (68–71%) (*Ribeiro et al., 2018*), questioning their utility for preventive purposes.

## Future implications

Global political alignments often shift, and the US military remains mobile in response, with a higher likelihood that a greater number of troops may be deployed to the Middle East in response to current trends. Engagement with local leishmania and malaria-endemic regions puts American servicemen and women at risk of disease contraction, thereby highlighting the disease burden in these foreign countries.

### Malaria

Malaria is endemic to much of the world, and the United States military will likely continue to engage in endemic areas. Drug-resistant strains of malaria have been found, indicating the need for further preventative measures (*Fukuda et al., 2011*; *Cui et al., 2015*). Most of these species are found in Asia (*White et al., 2014*). Globalization of the Asian continent as well as potential future military involvement in Asia could result in military exposure to such species, resulting in, at a minimum complicated treatment. For example, *P. vivax* is endemic to North Korea (*Nishiura et al., 2018*), and though present rapprochement between both countries and South Korea seems to have downplayed the threat of military engagement, US troops still engage in military runs in neighboring South Korea. This

puts US troops at risk of exposure if military forces press north. Furthermore, malaria was eradicated in South Korea in the 1970s, but soldiers (and increasingly, civilians as well) have been diagnosed with *P. vivax* malaria in the North-South demilitarized zone (*Lee et al., 2002*; *Im et al., 2018*), including reports of natural hybridization between local mosquito species and changing meteorological factors, to further perpetuate current observation (*Choochote et al., 2014*; *Phasomkusolsil et al., 2014*; *Chang et al., 2016*; *Hwang et al., 2016*).

### Leishmaniasis

CL is found primarily in the Middle East, and the US military continues to be at risk of infection. The *Leishmania* disease burden in the Middle East is estimated at 100,000 cases annually, indicating that exposure risk remains significant (*Salam, Al-Shaqha & Azzi, 2014*). Iran and Pakistan are of note in this region: both countries steadily trending towards increased prevalence, opposite of most other neighboring nations (*Rahman et al., 2010*; *Orsborne et al., 2016*). True infection rates may be greater than reported due to the fact that majority of affected rural communities lack the infrastructure for precise diagnosis and reporting, and the persistent antagonistic attitude to healthcare workers arising from many years of counterintelligence activities and breakdown of trust. Ongoing civil conflicts in Syria compounds the growing and untenable numbers of individuals symptomatic for disease or sub-clinically infected, showing up as refugees and present clinically in North America, Europe and other countries (*Koltas et al., 2014*; *Saroufim et al., 2014*; *Bradshaw & Litvinov, 2017*; *Wollina et al., 2018*; *Mockenhaupt et al., 2016*). This further compounds disease status among deployed servicemen and women returning to the West, igniting the need for expanding infectious disease experts into current healthcare system, as well as revising medical school curriculum to include training on 'exotic' tropical or subtropical diseases.

## CONCLUSIONS

Parasitic diseases are common worldwide, and exposure is common in most deployment locations of the United States military. The majority are not yet preventable with vaccines, and the treatments and prophylaxis that are available are accompanied by many side effects. It is important to continue funding for the treatment and eradication of infectious diseases worldwide for many reasons, including the exposure danger posed to the armed forces. Significantly, the possibility that these pathogens can be imported into the United States by returning service men and women, unknowingly serving as reservoir hosts, should serve as an alarm bell for us to re-evaluate, refocus and strategize how to face the challenges of the 21st century military deployment, protecting them from local pathogenic insults that can potentially lead to epidemics in the homeland.

## ACKNOWLEDGEMENTS

Bolaji Thomas is supported by a Research Laboratory and Faculty Development Award, College of Health Sciences and Technology, Rochester Institute of Technology. The views expressed are those of the authors and not necessarily those of the Rochester Institute of Technology.

### Funding

Bolaji Thomas is supported by a Research Laboratory and Faculty Development Award, College of Health Sciences and Technology, Rochester Institute of Technology. This work also received additional funding from Miller Chair in International Education (BNT). The funders had no role in study design, data collection and analysis, decision to publish, or preparation of the manuscript.

### Grant Disclosures

The following grant information was disclosed by the authors:
Research Laboratory and Faculty Development Award.
College of Health Sciences and Technology.
Rochester Institute of Technology.
Miller Chair in International Education (BNT).

### Competing Interests

The authors declare there are no competing interests.

### Author Contributions

- Kaylin J. Beiter performed the experiments, contributed reagents/materials/analysis tools, authored or reviewed drafts of the paper, approved the final draft.
- Zachariah J. Wentlent performed the experiments, authored or reviewed drafts of the paper, approved the final draft.
- Adrian R. Hamouda performed the experiments, approved the final draft.
- Bolaji N. Thomas conceived and designed the experiments, performed the experiments, contributed reagents/materials/analysis tools, authored or reviewed drafts of the paper, approved the final draft.

### Data Availability

This is a review paper. No raw code or raw data is involved.

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
