# Peer review of "Nonconventional opponents: a review of malaria and leishmaniasis among United States Armed Forces"

_PeerJ, doi:10.7717/peerj.6313_

## Round 0.1 · original submission · Major Revisions

While both Reviewers found your manuscript interesting and relevant, they have both raised a number of issues that need to be addressed.

Reviewer 1 ·

Basic reporting

Authors use the English language appropriately. Suggestions for modifying writing were minimal. The authors use several old and current references, adequate to the context of the study, but also suggest the inclusion of relevant works to complement some points of writing.

- Is the review of broad and cross-disciplinary interest and within the scope of the journal?
Yes.
- Has the field been reviewed recently?
No.
- If so, is there a good reason for this review (different point of view, accessible to a different audience, etc.)?
Yes.
- Does the Introduction adequately introduce the subject and make it clear who the audience is/what the motivation is?
Yes.

Experimental design

Article content is within the Aims and Scope of the Journal. Careful investigation with good writing. Methods need better description to replicate.

- Is the Survey Methodology consistent with a comprehensive, unbiased coverage of the subject?
More and less.
- If not, what is missing?
The authors can further detail the study methodology.
- Are sources adequately cited?
Yes.
- Quoted or paraphrased as appropriate?
Yes.
- Is the review organized logically into coherent paragraphs/subsections?
Yes.

Validity of the findings

Meaningful replication encouraged where rationale and benefit to literature is clearly stated. Conclusion are well stated, linked to original research question and limited to supporting results.

- Is there a well developed and supported argument that meets the goals set out in the Introduction?
Yes.
- Does the Conclusion identify unresolved questions / gaps / future directions?
Yes.

Additional comments

The manuscript presents a revision character and the results and conclusions contemplate the proposed objective. I make some considerations, however, to improve writing, especially some parts of the text.

- Abstract (lines 57 to 58): Visceral leishmaniasis has been extensively studied in the world, especially in endemic countries as Brazil, so I think it is pertinent for the authors to clarify that the disease is still little addressed in the US.

- Introduction (lines 84 to 86): The authors should tell when soldiers were negatively affected and cite sources consistent with the statement.

- Survey Methodology (lines 112 to 119): For this topic I make some suggestions. I believe that the authors can write in more detail what publications were considered for the review (if only original articles, common and/or systematic revisions, prior notes, brief communications, meta-analyzes, etc.). I also think it pertinent for the authors to cite the period of time considered for reading the references used (were articles sought from 1980 until today? Was a criterion of temporal selection of these studies used?). Another point that could be approached by the authors was the languages that these selected papers presented (English, Spanish, French, etc.) and the exclusion criteria of the researches that were not selected for review.

- Malaria (lines 125 to 128): The authors refer to a 2017 update that revealed 32 cases of malaria infection among US military personnel stationed overseas. How many soldiers there were among 32 cases? I find it interesting to situate the reader on this percentage of infection.

- Malaria (Causes) (lines 150 to 152): I suggest that the authors better explain this passage by highlighting that in the past, malaria was known as tertian or quartan fever, depending on the parasite species. This was related to the way in which the fever occurred in the patients after about 15 days: in the case of the tertian, the fever occurred every 48 hours (P. vivax, P. falciparum and P. ovale), and in the case of quartan, every 72 hours (P. malariae). The authors can cite references from malaria.

- Leishmania (Causes) (lines 225 to 228): In this topic I make two considerations. The authors can cite the metacyclic promastigote forms as infective for the vertebrate hosts during the blood meal of sandfly. The authors can also cite the diffuse cutaneous form as one of the clinical forms of tegumentary leishmaniasis.

- Leishmania (Causes) (lines 232 to 234): I suggest rewriting this sentence in other words. I considered the phrase confusing to my understanding.

- Leishmania (Causes) (lines 234 to 238): I suggest that the authors can cite more current US military infection data, since in the topic "Prevention and Treatment" (lines 261 to 262) the authors cite "Rowland 2015" as a reference in the passage "The reduced incidence rate of recent years has been attributed to better equipment and an emphasis on personal protective measures". What is this most current incidence rate?

- Leishmania (Symptons) (lines 241 to 242): This information is true: "Lesions can be self-resolving, as is often the case in many cases of CL", without treatment? I think that the authors can look for new references to better detail this statement.

- Leishmania (Prevention and Treatment) (lines 286): The authors refer to recent reports of treatment failures in South and Latin America leishmaniasis, but do not cite any reference to this consideration.

- Other: Military Canine Leishmaniasis (lines 312 to 314): The authors comment on a larger study volume in Mediterranean region and Brazil, however, could have cited some researches, including current studies of disease burden that have been published.

- Conclusions (line 376): The authors can remove the word “are” from italics.

·

Basic reporting

No comments

Experimental design

As strictly a literature review, the objectives are clear but the methods were not properly described. I have missed some of the most important and recent publications in both diseases and the searching method seems to be biased to more clinical than epidemiological publications.

Validity of the findings

No comments

Additional comments

This manuscript summarizes a literature search about malaria and leishmaniasis in the US military personnel. As strictly a literature review, the objectives are clear but the methods were not properly described. I have missed some of the most important and recent publications in both diseases and the searching method seems to be biased to more clinical than epidemiological publications. Overall, the paper is well-written and the results would be useful for future plans of improving malaria and leishmaniasis control.

Specific comments:

Line 57 - I will not use the term canine leishmaniasis since you are talking about human leishmaniasis caused by leishmania infantum, for which the main reservoir are dogs. The diseases in dogs is known as canine leishmaniasis in humans in some regions is known as zoonotic visceral leishmaniasis (ZVL) but can also be named as human visceral leishmaniasis caused by L. infantum.
Line 115 - The methodology should be more specific and include details about the searching methods. All the subject directed key words should be included. The list of words used must be limited otherwise the search will be impossible to manage. The word “etc” does not provide precise information. The methods must also include if two or three words were search at the same time and what link was used (AND/OR).
Line 161 – this reference is missing (Yohannes et al., 2016)
Line 181 - please add that has been observed in “rats”
Line 237 – Could you please add how the % of CL in British, Dutch and German soldiers compares with the % of CL in American soldiers?
Line 288 - Please add a reference
Line 291 - None of these 2 references is about the role of dogs as the main reservoir for leishmania infantum. Please include more appropriate references
Line 291-297 - Leishmania infantum is responsible for Zoonotic Visceral Leishmaniasis in humans and rarely induced cutaneous symptoms. Some of the references used here are not appropriate, please read these 2 publications and consider to introduced them in this section (Burza et al., 2018) (Quinnell and Courtenay, 2009)
Line 314 - Please add a reference here
Line 330 -331- Current Leishmania vaccines in dogs have been highly questioned as a prevention strategy since don’t stop transmission and only reduce clinical symptoms. Please include some precaution to this statement.

---

## Round 0.2 · accepted · Accept

Thank you for appropriately addressing the reviewers' comments.

# Reviewer 1 ·

Basic reporting

Is the review of broad and cross-disciplinary interest and within the scope of the journal?
Yes.

Has the field been reviewed recently? If so, is there a good reason for this review (different point of view, accessible to a different audience, etc.)?
Yes.

Does the Introduction adequately introduce the subject and make it clear who the audience is/what the motivation is?
Yes.

Experimental design

Is the Survey Methodology consistent with a comprehensive, unbiased coverage of the subject? If not, what is missing?
Yes.

Are sources adequately cited? Quoted or paraphrased as appropriate?
Yes.

Is the review organized logically into coherent paragraphs/subsections?
Yes.

Validity of the findings

Is there a well developed and supported argument that meets the goals set out in the Introduction?
Yes.

Does the Conclusion identify unresolved questions / gaps / future directions?
Yes.

·

Basic reporting

This review has improved after the revisions. Authors showed willingness to to adapt the manuscript the to the reviewers suggestion and the final outcome is an interest review in malaria and leishmania within the USA army

Experimental design

The content is within the scope of the journal and methods are described with sufficient detail

Validity of the findings

The benefit to the literature is clearly state. Conclusions link to the original questions

Additional comments

This is a nice review that is going to be appreciated by the researches in the area of malaria and leishmania